# REGEN: Reference-Guided Synthetic Multivariate Time Series Generation for Forecasting

Moulik Gupta [1]   Aaditya Jain [1]   Dhruv Kumar [1 2]   Murari Mandal [1 3 †]   Saurabh Deshpande [1 †]

## Abstract

Time series foundation models are increasingly pretrained on synthetic data, yet existing generators are largely designed for univariate settings or rely on domain-agnostic priors that fail to capture the periodic structure and cross-variable coupling characteristic of real multivariate corpora. We propose REGEN, a reference-guided pipeline that treats a small number of real multivariate sequences as structural scaffolds for synthesis. REGEN decomposes each reference into a phase-aligned periodic template, per-variable stochastic residuals modeled via deep kernel learning with a Gaussian-process prior, and cross-variable dependencies encoded as a directed acyclic graph with fitted lagged couplings. Because every component is estimated from real observations, the synthetic corpus preserves domain-specific multivariate structure while generating novel trajectories at arbitrary scale. When used to pretrain Moirai-small from scratch, REGEN reduces zero-shot MSE by 41% over TimeGAN and by 2.3% over CauKer. Across twelve datasets spanning energy, traffic, climate, and cloud infrastructure, models trained on REGEN data approach real-data baselines, showing that reference-guided multivariate synthesis is a strong inductive bias for foundation model training.

## 1. Introduction

Multivariate time series forecasting underpins applications ranging from energy management and traffic control to climate modeling and cloud operations (Cai et al., 2024; Han et al., 2024; Shao et al., 2022). Yet progress is often limited by data scarcity: many target domains contain only a few observed sequences, making reliable generalization difficult (Taga et al., 2025). This scarcity is especially acute for foundation model pretraining, where synthetic data has become a core component of training pipelines (Ansari et al., 2024; Das et al., 2024; Auer et al., 2026). Unlike text or vision, time series data is fragmented across domains, often restricted, and expensive to collect. This raises a practical question: *can we generate synthetic multivariate time series faithful enough to a target domain to substitute for real training data?*

Existing approaches fall short for different reasons. Prior-based generators (Dooley et al., 2023; Ansari et al., 2024; Oreshkin et al., 2026) synthesize from mathematical primitives with no domain grounding. Recent multivariate variants, including (Taga et al., 2025) and (Xie et al., 2025), add GP-based structure and causal graphs, but these dependencies are still sampled from generic priors rather than the target domain. Data-driven generators such as (Yoon et al., 2019) learn from real sequences but require large collections and offer limited control over periodic structure or cross-variable coupling. No existing method is designed for conditioning on only a few real multivariate references to produce domain-faithful training corpora.

We instead treat scarce real sequences as structural scaffolds for synthesis. Our proposed method, REGEN, decomposes each reference into three interpretable components: a phase-aligned periodic template, stochastic residuals modeled with deep kernel learning and a Gaussian-process prior (Duvenaud et al., 2013), and a structural causal model (Bongers et al., 2021) implemented as a directed acyclic graph with fitted lagged couplings. Because each component is estimated from real observations, the synthetic corpus preserves domain-specific periodic morphology, local uncertainty, and cross-variable dependence while generating novel trajectories at arbitrary scale. We evaluate REGEN across twelve datasets and three forecasting backbones, and show that it consistently outperforms existing synthetic generators for both per-domain training and foundation model pretraining.

---

[†]Equal supervision.   [1]Birla AI Labs, Office of Ananya Birla [2]Birla Institute of Technology and Science, Pilani [3]Kalinga Institute of Industrial Technology, Bhubaneswar. Correspondence to: {moulik.gupta-c, aaditya.jain, dhruv.kumar-c, murari.mandal-c, saurabh.deshpande-c}@oab.adityabirla.com.

*Proceedings of the $43^{rd}$ International Conference on Machine Learning*, Seoul, South Korea. PMLR 306, 2026. Copyright 2026 by the author(s).

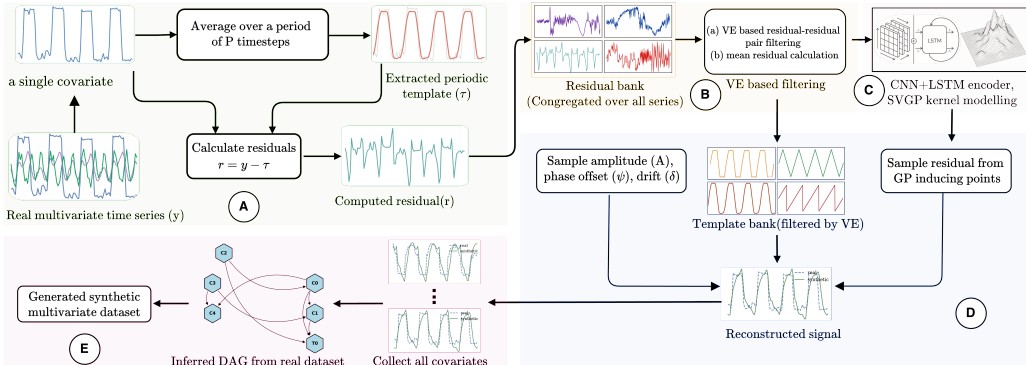

*Figure 1.* REGEN pipeline. **A:** Extract a phase-aligned periodic template and compute residuals from the real multivariate time series. **B:** Aggregate residuals across series and apply VE-based filtering to retain reliable template–residual structure. **C:** Fit a CNN+LSTM encoder with an SVGP-based deep kernel prior to model residual dynamics. **D:** Sample template parameters and GP residuals, then combine them to reconstruct synthetic signals. **E:** Use the inferred DAG to inject cross-variate dependencies and assemble the final synthetic multivariate dataset.

## 2. Related Work

Data-driven generators such as TimeGAN (Yoon et al., 2019), C-RNN-GAN (Mogren, 2016), and RCGAN (Esteban et al., 2017) produce realistic sequences but operate as black boxes with no explicit control over periodic structure or cross-variable coupling, and require sufficient real data to train. Prior-based generators instead synthesize from domain-agnostic primitives: ForecastPFN (Dooley et al., 2023) uses Bayesian priors; Chronos composes GP kernels for univariate series (Ansari et al., 2024); TimePFN extends this to multivariate settings (Taga et al., 2025); CauKer adds a causal DAG for coherent cross-variable structure (Xie et al., 2025); and SarSim uses SARIMA-based simulation for large-scale pretraining (Oreshkin et al., 2026). While suited to foundation model pretraining, these methods are domain-agnostic and unsuitable for target-conditioned augmentation.

## 3. REGEN

Let $\mathbf{x}_{1:T}^{(m)} \in \mathbb{R}^{C \times T}$ denote one reference multivariate series, where $m$ indexes the available references, $c \in \{1, \dots, C\}$ indexes covariates, and $x_{c,t}^{(m)}$ denotes the value of covariate $c$ at time step $t$. The key design choice in REGEN is to treat real observations as a structural scaffold from which we explicitly extract reusable periodic morphology and stochastic variation.

**Reference decomposition.** For each covariate $c$, we first standardize the full trajectory across time and then decompose the resulting sequence into a periodic template and a residual process. This separation between recurring structure and stochastic remainder is directly inspired by classical seasonal decomposition and structured probabilistic modeling of temporal signals (Cleveland et al., 1990; Roberts et al., 2013). Let $z_{c,t}$ denote the normalized value at time

step $t$ for covariate $c$. We set the dominant period $P_c$ from domain knowledge when the periodicity is known a priori (See Appendix A.2). The template is then obtained by phase-aligned averaging:

$$\tau_c(p) = \frac{1}{|\mathcal{I}_{c,p}|} \sum_{t \in \mathcal{I}_{c,p}} z_{c,t}, \quad \mathcal{I}_{c,p} = \{t : t \equiv p \pmod{P_c}\}.$$

We then define the residual as $r_{c,t} = z_{c,t} - \tau_c(t \bmod P_c)$. In the multi-reference setting, this procedure yields a bank of reference-specific templates $\{\tau_c^{(m)}\}_{m=1}^M$. We retain these templates as interchangeable structural motifs during synthesis, while fitting the stochastic model on the residual behavior of the retained references. This decomposition is the central inductive bias of REGEN: periodic structure is borrowed directly from the target domain, whereas novelty is pushed into the residual path, in the same spirit as decomposition-based time-series modeling (Cleveland et al., 1990; Roberts et al., 2013).

**Residual dynamics via DKL.** After removing the template, the remaining residual sequence captures local variability, irregular fluctuations, and short-range dependencies. We model this component with deep kernel learning (Wilson et al., 2016) and a Gaussian-process prior over latent temporal features (Rasmussen & Williams, 2006). A neural encoder $f_\theta$ maps a rolling history window of length $W$ to a latent state,

$$h_{c,t} = f_\theta\big(r_{c,t-W:t-1}\big),$$
$$g_c \sim \mathcal{GP}(0, k_c),$$
$$r_{c,t} = g_c(h_{c,t}) + \epsilon_t,$$

with $\epsilon_t \sim \mathcal{N}(0, \sigma_{\epsilon,c}^2)$. In our implementation, the encoder is realized with a recurrent history summarizer in the spirit of LSTM sequence models (Hochreiter & Schmidhuber, 1997).

**Synthetic trajectory generation.** To sample a new syn-

thetic series, we first choose a trajectory-specific template $\tau_c^{(m)}$ for each covariate, then draw an amplitude factor $a_c$ and a low-frequency sinusoidal drift process $d_{c,t}$. Residuals are generated autoregressively from the predictive distribution of the DKL model,

$$\tilde{r}_{c,t} = \hat{\mu}_{c,t} + \eta_c \hat{\sigma}_{c,t}\xi_t, \qquad \xi_t \sim \mathcal{N}(0,1),$$

where $(\hat{\mu}_{c,t}, \hat{\sigma}_{c,t}^2)$ are the GP predictive moments and $\eta_c$ is a temperature parameter controlling diversity. The final normalized synthetic signal is reconstructed as

$$\tilde{z}_{c,t} = \bar{\tau}_c^{(m)} + a_c\Big(\tau_c^{(m)}\big(t \bmod P_c\big) - \bar{\tau}_c^{(m)}\Big)$$
$$+ \tilde{r}_{c,t} + d_{c,t},$$

followed by denormalization $\tilde{x}_{c,t} = \sigma_c \tilde{z}_{c,t} + \mu_c$. Scaling around the template mean $\bar{\tau}_c^{(m)}$ preserves the reference level while allowing the periodic amplitude to vary across synthetic draws. As a result, two samples can share the same coarse morphology while differing in volatility and residual evolution.

**Multivariate coupling.** We next apply a DAG-guided lag-mixing stage to couple the per-covariate synthetic trajectories into a coherent multivariate sample. This design follows the structural-causal perspective of directed graphical models (Peters et al., 2017) and is closely aligned with recent graph-based synthetic time-series generation work (Xie et al., 2025). We obtain the DAG by running causal discovery methods such as (Runge et al., 2019; Pamfil et al., 2020; Hyv"arinen et al., 2010) on the real dataset (See Appendix C). Let $\mathrm{Pa}(c)$ denote the parents of covariate $c$ in this directed acyclic graph and let $\mathcal{L}_{p\rightarrow c}$ be the set of admissible lags on edge $p \rightarrow c$. For a child variable, we form a parent contribution

$$u_{c,t} = \sum_{p\in\mathrm{Pa}(c)} \sum_{\ell\in\mathcal{L}_{p\rightarrow c}} w_{p,c,\ell}\, s_{p,t-\ell},$$
$$s'_{c,t} = \alpha_c s_{c,t} + (1-\alpha_c)u_{c,t},$$

where $s_{c,t}$ denotes either the intrinsic residual or the intrinsic full signal, depending on the mixing mode, and $\alpha_c \in (0,1)$ preserves covariate-specific dynamics (Peters et al., 2017; Xie et al., 2025).

## 4. Experiments and Results

**Experimental setup.** We evaluate iTransformer (Liu et al., 2024), S-Mamba (Wang et al., 2025), and DLinear (Zeng et al., 2023). These models span transformer, state-space, and linear families, with roughly 1M, 1.1M, and 4.7K parameters, respectively. All runs use an input context length of 96 and a prediction horizon of 24. For each transfer setting, the synthetic training set is matched in size to the corresponding real dataset, and in TRTR (Train on Real,

**Table 1:** Zero-shot forecasting transfer across sibling dataset pairs under **TRTR** and **TSTR** protocols, evaluated on three backbones. Each TSTR row directly below shows REGEN synthetic performance. The $\Delta$ (%) column reports relative change $(\mathrm{TSTR} - \mathrm{TRTR})/\mathrm{TRTR} \times 100$ in MSE.

| Pair | Direction | iTransformer | | | DLinear | | | SMamba | | |
|---|---|---|---|---|---|---|---|---|---|---|
| | | MSE | MAE | Δ(%) | MSE | MAE | Δ(%) | MSE | MAE | Δ(%) |
| BDG-2 Bear Panther | A→B | 0.32 | 0.36 | — | 0.29 | 0.36 | — | 0.29 | 0.34 | — |
| | D→B | 0.36 | 0.38 | +12.5 | 0.30 | 0.36 | +3.4 | 0.29 | 0.35 | 0.0 |
| | B→A | 0.41 | 0.40 | — | 0.41 | 0.39 | — | 0.42 | 0.34 | — |
| | C→A | 0.41 | 0.43 | 0.0 | 0.43 | 0.43 | +4.9 | 0.40 | 0.33 | −4.8 |
| BDG-2 Bull Hog | A→B | 0.50 | 0.49 | — | 0.48 | 0.45 | — | 0.49 | 0.46 | — |
| | D→B | 0.50 | 0.48 | 0.0 | 0.49 | 0.46 | +2.1 | 0.48 | 0.45 | −2.0 |
| | B→A | 0.33 | 0.40 | — | 0.36 | 0.39 | — | 0.33 | 0.40 | — |
| | C→A | 0.37 | 0.40 | +12.1 | 0.31 | 0.38 | −13.9 | 0.35 | 0.41 | +6.1 |
| Azure VM / Borg 2011 | A→B | 0.58 | 0.49 | — | 0.59 | 0.54 | — | 0.58 | 0.53 | — |
| | D→B | 0.59 | 0.50 | +1.7 | 0.59 | 0.52 | 0.0 | 0.61 | 0.55 | +5.2 |
| | B→A | 0.88 | 0.41 | — | 0.90 | 0.44 | — | 0.90 | 0.41 | — |
| | C→A | 0.90 | 0.45 | +2.3 | 0.92 | 0.42 | +2.2 | 0.97 | 0.42 | +7.8 |
| PEMS-04 / PEMS-08 | A→B | 0.32 | 0.34 | — | 0.30 | 0.29 | — | 0.29 | 0.30 | — |
| | D→B | 0.30 | 0.29 | −6.3 | 0.28 | 0.28 | −6.7 | 0.26 | 0.30 | −10.3 |
| | B→A | 0.31 | 0.31 | — | 0.32 | 0.31 | — | 0.30 | 0.28 | — |
| | C→A | 0.37 | 0.38 | +19.4 | 0.29 | 0.30 | −9.4 | 0.29 | 0.33 | −3.3 |
| Subseasonal / Precip. | A→B | 1.09 | 0.77 | — | 0.83 | 0.60 | — | 0.76 | 0.57 | — |
| | D→B | 1.01 | 0.73 | −7.3 | 0.90 | 0.64 | +8.4 | 0.74 | 0.56 | −2.6 |
| | B→A | 0.42 | 0.46 | — | 0.40 | 0.47 | — | 1.34 | 1.02 | — |
| | C→A | 0.40 | 0.43 | −4.8 | 0.35 | 0.39 | −12.5 | 1.51 | 1.10 | +12.7 |
| Res. PV / Load | A→B | 0.60 | 0.42 | — | 0.55 | 0.40 | — | 0.54 | 0.37 | — |
| | D→B | 0.54 | 0.38 | −10.0 | 0.64 | 0.41 | +16.4 | 0.54 | 0.37 | 0.0 |
| | B→A | 0.25 | 0.22 | — | 0.25 | 0.24 | — | 0.23 | 0.20 | — |
| | C→A | 0.26 | 0.20 | +4.0 | 0.31 | 0.29 | +24.0 | 0.22 | 0.23 | −4.3 |

**Colour key**: dark green TSTR beats TRTR by $> 3\%$; light green TSTR better by $\leq 3\%$; yellow within $\pm 3\%$ tolerance; red gap $> 3\%$. **Run key**: $A, B$ = real datasets; $C$ = synthetic from $A$; $D$ = synthetic from $B$. $A \rightarrow B$, $B \rightarrow A$ = TRTR; $D \rightarrow B$, $C \rightarrow A$ = TSTR.

Test on Real)/TRSTR (Train on the union of Real and Synthetic, Test on Real)/ TSTR (Train on Synthetic, Test on Real) we sample residual temperatures from $\mathcal{U}(1.2, 2.4)$ so the synthetic data covers plausible regions where the real dataset is rarer.

**Q1: Reference-guided transfer as a drop-in replacement for real sibling data.** We report TRTR and TSTR performance across all sibling pairs and backbones in Table 1. Roughly two-thirds of TSTR metric cells stay within the $\pm 3\%$ relative band of, or improve on, their TRTR baseline, meaning that in a large fraction of settings replacing real sibling data with REGEN-generated series costs little in forecast accuracy and is often competitive with using real sibling data directly.

*Domain-wise variation.* Traffic is the strongest domain. On the PEMS-04 / PEMS-08 pair, TSTR reduces MSE by 6.3%, 6.7%, and 10.3% across iTransformer, DLinear, and S-Mamba in one transfer direction, and remains better for DLinear and S-Mamba in the reverse direction. Its largest

*Table 2.* TRTR vs. TRSTR across twelve datasets and three back-bones. TRSTR adds a size-matched REGEN synthetic corpus to the real training pool. Augmentation improves or matches TRTR in the majority of settings. Green indicates improvement; red indicates degradation. Lower is better ($\downarrow$).

| Dataset | iTransformer | | | | DLinear | | | | S-Mamba | | | |
|---|---|---|---|---|---|---|---|---|---|---|---|---|
| | TRTR | | TRSTR | | TRTR | | TRSTR | | TRTR | | TRSTR | |
| | MSE | MAE | MSE | MAE | MSE | MAE | MSE | MAE | MSE | MAE | MSE | MAE |
| Bear | 0.41 | 0.40 | 0.38 | 0.40 | 0.41 | 0.39 | 0.42 | 0.43 | 0.42 | 0.34 | 0.42 | 0.34 |
| Panther | 0.32 | 0.36 | 0.27 | 0.35 | 0.29 | 0.36 | 0.29 | 0.34 | 0.29 | 0.34 | 0.29 | 0.33 |
| Azure VM | 0.88 | 0.41 | 0.78 | 0.30 | 0.90 | 0.44 | 0.88 | 0.42 | 0.90 | 0.41 | 0.87 | 0.40 |
| Borg | 0.58 | 0.49 | 0.55 | 0.40 | 0.59 | 0.54 | 0.55 | 0.54 | 0.58 | 0.53 | 0.54 | 0.50 |
| PEMS-04 | 0.31 | 0.31 | 0.25 | 0.27 | 0.32 | 0.31 | 0.30 | 0.31 | 0.30 | 0.28 | 0.28 | 0.29 |
| PEMS-08 | 0.32 | 0.34 | 0.27 | 0.29 | 0.30 | 0.29 | 0.30 | 0.28 | 0.29 | 0.30 | 0.26 | 0.29 |
| Bull | 0.33 | 0.40 | 0.30 | 0.35 | 0.36 | 0.39 | 0.32 | 0.35 | 0.33 | 0.40 | 0.32 | 0.40 |
| Hog | 0.50 | 0.49 | 0.47 | 0.42 | 0.48 | 0.45 | 0.47 | 0.44 | 0.49 | 0.46 | 0.43 | 0.45 |
| Subseq. | 0.42 | 0.46 | 0.43 | 0.47 | 0.40 | 0.47 | 0.44 | 0.48 | 1.34 | 1.02 | 1.08 | 0.73 |
| Subseq. Prec. | 1.09 | 0.77 | 0.92 | 0.72 | 0.83 | 0.60 | 0.79 | 0.62 | 0.76 | 0.57 | 0.78 | 0.57 |
| Res. PV | 0.25 | 0.22 | 0.24 | 0.23 | 0.25 | 0.24 | 0.31 | 0.28 | 0.23 | 0.20 | 0.29 | 0.26 |
| Res. Load | 0.60 | 0.42 | 0.54 | 0.33 | 0.55 | 0.40 | 0.63 | 0.43 | 0.54 | 0.37 | 0.54 | 0.37 |

*Table 3.* Dataset-wise iTransformer comparison. The TRTR block is shown as a neutral real-data reference. In the TSTR columns, green marks the lower-error synthetic method between TimeGAN and REGEN for each metric; lower is better.

| Dataset | TRTR | | TimeGAN | | REGEN (Ours) | |
|---|---|---|---|---|---|---|
| | MSE | MAE | MSE | MAE | MSE | MAE |
| Bear | 0.41 | 0.40 | 0.63 | 0.50 | 0.41 | 0.43 |
| Panther | 0.32 | 0.36 | 0.40 | 0.45 | 0.36 | 0.38 |
| Azure VM | 0.88 | 0.41 | 1.04 | 0.51 | 0.90 | 0.45 |
| Borg | 0.58 | 0.49 | 0.85 | 0.69 | 0.59 | 0.50 |
| PEMS-04 | 0.31 | 0.31 | 0.45 | 0.45 | 0.37 | 0.38 |
| PEMS-08 | 0.32 | 0.34 | 0.40 | 0.38 | 0.30 | 0.29 |
| Bull | 0.33 | 0.40 | 0.32 | 0.36 | 0.37 | 0.40 |
| Hog | 0.50 | 0.49 | 0.47 | 0.49 | 0.50 | 0.48 |
| Subseasonal | 0.42 | 0.46 | 0.42 | 0.45 | 0.40 | 0.43 |
| Sub. precip. | 1.09 | 0.77 | 1.11 | 0.80 | 1.01 | 0.73 |
| Residential PV | 0.25 | 0.22 | 0.27 | 0.20 | 0.26 | 0.20 |
| Residential Load | 0.60 | 0.42 | 0.71 | 0.49 | 0.54 | 0.38 |

gap is the DLinear TSTR setting at +16.4% MSE, and the appendix t-SNE diagnostic (Figure 2) shows Residential PV as the clearest geometric exception.

**Q2: Synthetic augmentation improves over real-only training.** We compare TRTR against TRSTR, training on the union of the real corpus and a size-matched synthetic corpus, across the same twelve datasets and three backbones in Table 2.

*Augmentation helps broadly, but the effect is architecture-dependent.* For iTransformer and S-Mamba, TRSTR usually matches or beats TRTR across the majority of datasets: both backbones benefit from the increased effective diversity of the combined corpus, absorbing the additional synthetic variation rather than overfitting the original distribution. DLinear shows the weakest and most mixed gains.

*The most practically significant augmentation gains appear where datasets are small or structurally difficult.* Subseasonal S-Mamba improves from an MSE of 1.34 (TRTR) to 1.08 (TRSTR), a 19.4% reduction, even though iTrans-

*Table 4.* Full-corpus TSTR evaluation with Moirai-small, comparing the different synthetic corpora used for pre-training. Lower MSE, MAE, MASE, and WQL are better. Best results in green.

| Synthetic Corpus | MSE | MAE | MASE | WQL |
|---|---|---|---|---|
| TimeGAN | 392.46 | 8.38 | 1.45 | 0.23 |
| CauKer | 236.7 | 6.64 | 1.11 | 0.17 |
| REGEN (Ours) | 231.14 | 5.86 | 0.98 | 0.17 |

former and DLinear do not improve on that dataset. Azure VM improves across all three backbones, and Hog also improves consistently across all three, suggesting that the synthetic corpus contributes rare regimes and cross-variate patterns that are underrepresented in the finite real training set. The augmentation results suggest that synthetic data complements rather than replays the observed distribution, covering plausible low-density regions.

**Q3: REGEN provides stronger training signal than existing alternatives.** We compare REGEN with TimeGAN (Yoon et al., 2019), a data-driven adversarial generator trained on observed data, and CauKer (Xie et al., 2025), a reference-free prior-based generator whose DAG and kernel bank are drawn from domain-agnostic priors. Because both (TimeGAN and REGEN) methods use the same observed data and synthetic-corpus budget, performance gaps reflect synthetic-data quality rather than volume. In the dataset-wise iTransformer comparison of Table 3, REGEN achieves lower MSE and MAE on 10 of 12 datasets.

REGEN *vs CauKer vs TimeGAN.* As CauKer is domain-agnostic, per-dataset TSTR comparisons are not well matched to its design. We therefore pool all synthetic datasets from all three generators, pre-train Moirai-small from scratch on each pooled corpus, and evaluate zero-shot on the pooled real benchmark. Even in this CauKer-favorable setting, REGEN reduces MSE by 2.3% relative to CauKer (Table: 4).

## 5. Conclusion

We have shown that reference-guided synthetic generation is a practical strategy for constructing multivariate training corpora for time series foundation models. By conditioning on a small number of real sequences, REGEN preserves domain-specific periodic structure, calibrated residual dynamics, and cross-variable dependence. Foundation models pretrained on REGEN corpora outperform those trained on both data-driven and prior-based alternatives, while per-domain evaluations confirm that the generated data approaches real-data utility across diverse domains. Our evaluation is currently limited to Moirai-small for foundation model pretraining and iTransformer for component ablations; SCM-based mixing also shows non-uniform benefits in low-data settings. We discuss these limitations and future directions in Appendix B.

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

## A. Dataset and System Details

*Table 5.* Twelve real-world datasets used in evaluation, spanning energy, cloud, traffic, and climate, with summary statistics for sampling frequency, characteristic template period $P$, series count, targets, and covariates.

| Dataset | Source | Domain | Frequency | Template $P$ | # Time Series | # Targets | # Covariates |
|---|---|---|---|---|---|---|---|
| BDG-2 Bear | BuildingsBench (Emami et al., 2023) | Energy | H | 24 | 91 | 1 | 0 |
| BDG-2 Panther | BuildingsBench (Emami et al., 2023) | Energy | H | 24 | 105 | 1 | 0 |
| Azure VM Traces 2017 | CloudOpsTSF (Woo et al., 2023) | Cloud | 5T | 288 | 10,000 | 1 | 2 |
| Borg Cluster Data 2011 | CloudOpsTSF (Woo et al., 2023) | Cloud | 5T | 288 | 10,000 | 2 | 5 |
| PEMS-04 | LibCity (Jiang et al., 2023) | Traffic | 5T | 288 | 307 | 3 | 0 |
| PEMS-08 | LibCity (Jiang et al., 2023) | Traffic | 5T | 288 | 170 | 3 | 0 |
| BDG-2 Bull | BuildingsBench (Emami et al., 2023) | Energy | H | 24 | 41 | 1 | 3 |
| BDG-2 Hog | BuildingsBench (Emami et al., 2023) | Energy | H | 24 | 24 | 1 | 5 |
| Subseasonal | SubseasonalClimateUSA (Mouatadid et al., 2023) | Climate | D | 365 | 862 | 4 | 0 |
| Subseasonal Precipitation | SubseasonalClimateUSA (Mouatadid et al., 2023) | Climate | D | 365 | 862 | 1 | 0 |
| Residential PV Power | LOTSA_Others (Woo et al., 2024) | Energy | T | 1440 | 233 | 3 | 0 |
| Residential Load Power | LOTSA_Others (Woo et al., 2024) | Energy | T | 1440 | 271 | 3 | 0 |

### A.1. Experimental System Details

All experiments reported in the main paper and supplementary material were run on the same machine. The hardware consisted of an NVIDIA A10G GPU with 22.1 GB of VRAM, an AMD EPYC 7R32 CPU with 4 cores and 8 threads, 32 GB of system RAM, and a 242 GB SSD. The software environment used Ubuntu 24.04.4 LTS. These specifications cover both synthetic-data generation and downstream forecasting experiments.

### A.2. Dataset Details

**Characteristic Period.** $P$ denotes the characteristic period length, in time steps, used for phase alignment in the template-extraction stage. We use one full daily cycle for the sub-daily datasets and one climatological yearly cycle for the daily climate datasets, so the frequency codes map as follows: H $\mapsto P = 24$, 5T $\mapsto P = 288$, T $\mapsto P = 1440$, and D $\mapsto P = 365$.

The 12 datasets in Table 5 are grouped into six source-matched pairs so that each pair shares a common collection setting or application domain while still differing in scale, dimensionality, or predictand. We briefly summarize those six pairings here because they anchor the cross-dataset comparisons used throughout the main paper.

**BuildingsBench: Bear and Panther.** Bear and Panther are hourly building-energy datasets from BuildingsBench, both focused on single-target load forecasting without additional covariates. They form a clean univariate pair for evaluating whether a method can transfer across buildings that share broad consumption rhythms but differ in occupancy patterns, control policies, and building-specific demand variability.

**CloudOpsTSF: Azure VM Traces 2017 and Borg Cluster Data 2011.** Azure VM Traces 2017 and Borg Cluster Data 2011 represent cloud-resource monitoring at 5-minute resolution. This pair is useful because both datasets capture operational infrastructure telemetry, but Azure is a simpler single-target setting with two covariates, whereas Borg is multivariate and more heterogeneous, making the pair a direct test of how methods scale from lighter to richer cloud traces.

**LibCity: PEMS-04 and PEMS-08.** PEMS-04 and PEMS-08 are traffic datasets from LibCity with 5-minute sampling and three target channels per sensor. They provide a matched transportation pair in which both tasks exhibit strong daily and weekly rhythms, while differing in network size and sensor topology, so they probe whether a model preserves structured periodicity under varying spatial scales.

**BuildingsBench: Bull and Hog.** Bull and Hog return to BuildingsBench, but now in covariate-rich hourly settings rather than the univariate Bear/Panther case. Because both datasets model building demand with exogenous drivers, this pair helps separate performance gains from simple periodic load reconstruction versus the harder problem of handling auxiliary variables that may shift or weaken the dominant seasonal pattern.

**SubseasonalClimateUSA: Subseasonal and Subseasonal Precipitation.** Subseasonal and Subseasonal Precipitation come from the same climate benchmark and both aggregate daily measurements across 862 series, but they differ sharply in target dimensionality. The pair therefore isolates how a method behaves when the broader meteorological setting is held fixed while the predictive task changes from a four-target subseasonal forecasting problem to a precipitation-only version with a narrower signal profile.

**LOTSA Others: Residential PV Power and Residential Load Power.** Residential PV Power and Residential Load Power are minute-level energy datasets drawn from the LOTSA collection. They form a natural household-energy pair because both reflect residential behavior at fine temporal resolution, yet PV generation is dominated by solar forcing while load reflects human activity and appliance usage, giving a useful contrast between externally driven and behavior-driven dynamics.

## B. Limitations and Future Work

Our current evaluation has three main limitations. First, due to computational constraints, we report full-corpus foundation model pretraining results for Moirai-small only; extending to additional architectures would require training each model from scratch on each synthetic corpus, which is a natural and important next step. Second, our component ablations are reported only for iTransformer, so while they support the design rationale, they do not yet establish that the same component-level effects hold across all downstream model families. Third, SCM-based cross-variable mixing is not uniformly beneficial: in low-data, higher-dimensional settings such as Hog, graph estimation becomes noisy and the induced coupling can hurt performance.

## C. Consensus DAG Estimation

The graph $\mathcal{G}$ used in the SCM mixing stage is estimated directly from the real multivariate data through a consensus causal-discovery pipeline. We avoid relying on a single discovery routine because different estimators emphasize different dependence structures and can behave differently across datasets. Instead, we infer candidate edges with four complementary procedures: PCMCI (Runge et al., 2019), DYNOTEARS (Pamfil et al., 2020), VAR-LiNGAM (Hyv"arinen et al., 2010), and a non-linear Granger-style test based on random-forest feature importance (Breiman, 2001). The final graph retains only edges that recur across the dataset and receive support from multiple methods.

**Per-series preprocessing.** For each real series $s$, we stack all available variates into a single multivariate trajectory

$$\mathbf{z}_t^{(s)} = \big(z_{1,t}^{(s)}, \ldots, z_{C,t}^{(s)}\big)^{\top}, \qquad t = 1, \ldots, T_s, \quad (1)$$

placing dynamic covariates before the target variates. Missing values are imputed by linear interpolation, with forward/backward filling at the boundaries when needed. Each variate is then standardized independently,

$$\widetilde{z}_{c,t}^{(s)} = \frac{z_{c,t}^{(s)} - \mu_c^{(s)}}{\sigma_c^{(s)} + \varepsilon}, \qquad (2)$$

so that discovery is driven by temporal dependence rather than raw scale. Here $C$ is the total number of variates in the multivariate series and $T_s$ is its observed length.

**Candidate edge discovery.** We run all four discovery methods on each standardized series over a fixed lag window $\ell \in \{0, 1, \ldots, L_{\max}\}$. For a given series and method, a candidate edge is indexed by an ordered triple $(u, v, \ell)$, where $u \in \{1, \ldots, C\}$ is the source variate, $v \in \{1, \ldots, C\}$ is the destination variate, and $\ell$ is the lag relating source time $t - \ell$ to destination time $t$. Thus, $\ell = 0$ denotes an instantaneous or contemporaneous relation, whereas $\ell > 0$ denotes

a delayed relation. Every method returns a set of candidate relations of the form

$$\widetilde{z}_{u,t-\ell}^{(s)} \to \widetilde{z}_{v,t}^{(s)} \qquad (3)$$

together with a raw edge-strength score $a_{u,v,\ell}^{(s,m)}$, where $m \in \{1, \ldots, 4\}$ indexes the discovery method. We keep both contemporaneous and positive-lag edges at this stage. However, self-edges with $u = v$ are removed before the final consensus graph is constructed. The reason is architectural rather than causal: the periodic template stage already carries the dominant self-lag or autoregressive structure, so re-inserting self-links into the SCM mixing graph would double-count that dependence, degrade signal quality, and typically reduce fidelity rather than improve it.

**Aggregation across series.** The raw scores produced by different algorithms are not directly comparable, so we normalize them within each method and series before aggregation. Let $\mathcal{E}^{(s,m)}$ denote the set of candidate edges returned by method $m$ on series $s$. We define the normalized score

$$\widehat{a}_{u,v,\ell}^{(s,m)} = \begin{cases} \dfrac{\left| a_{u,v,\ell}^{(s,m)} \right|}{\max\limits_{(i,j,r) \in \mathcal{E}^{(s,m)}} \left| a_{i,j,r}^{(s,m)} \right|}, & (u, v, \ell) \in \mathcal{E}^{(s,m)}, \\ 0, & \text{otherwise.} \end{cases} \qquad (4)$$

where $(i, j, r)$ is simply a dummy edge index ranging over all candidate source variates $i$, destination variates $j$, and lags $r$ returned by method $m$ on series $s$. This rescales all detected edges for a given method/series pair into $[0, 1]$ while preserving their relative ordering. We then aggregate each edge across the $S$ real series through two summaries. First, its frequency of occurrence under method $m$ is

$$f_{u,v,\ell}^{(m)} = \frac{1}{S} \sum_{s=1}^{S} \mathbf{1}\Big[ (u, v, \ell) \in \mathcal{E}^{(s,m)} \Big]. \qquad (5)$$

Second, its mean normalized strength over the series in which it appears is

$$\bar{a}_{u,v,\ell}^{(m)} = \frac{\sum_{s=1}^{S} \widehat{a}_{u,v,\ell}^{(s,m)}}{\max\Big( 1, \sum_{s=1}^{S} \mathbf{1}\big[ (u, v, \ell) \in \mathcal{E}^{(s,m)} \big] \Big)}. \qquad (6)$$

These two quantities separate stability across series from within-method edge magnitude, yielding a dataset-level summary for each method rather than a separate graph for every series. At this point the edge-lag triple $(u, v, \ell)$ is still represented by *method-specific* summaries $\bar{a}_{u,v,\ell}^{(m)}$ rather than by a single pooled score. In other words, after this step there can still be up to four aggregated strengths for the same $(u, v, \ell)$, one for each discovery method.

**Consensus graph construction.** A vote is counted for an edge only after two levels of filtering. First, a particular

discovery method $m$ casts a vote for $(u, v, \ell)$ only if that relation appears in more than a threshold fraction of the dataset. In our implementation, this per-method support condition is

$$f^{(m)}_{u,v,\ell} > \tau_{\text{freq}}, \qquad \tau_{\text{freq}} = 0.2. \tag{7}$$

Equivalently, the edge must appear in more than 20% of the series for that method. Defining the per-method vote indicator as

$$b^{(m)}_{u,v,\ell} = \mathbf{1}\left[ f^{(m)}_{u,v,\ell} > \tau_{\text{freq}} \right], \tag{8}$$

the second requirement is cross-method agreement: an edge is kept only if at least two of the four discovery methods vote for it,

$$\sum_{m=1}^{4} b^{(m)}_{u,v,\ell} \geq 2. \tag{9}$$

Thus, an edge enters the final consensus graph only if *both* conditions hold: (a) for a given method it appears in more than $\tau_{\text{freq}}$ of the dataset, and (b) at least two methods vote for that same edge. For edges that survive this voting stage, we then pool the method-specific strengths into a single final consensus score,

$$w_{u,v,\ell} = \frac{\sum_{m=1}^{4} b^{(m)}_{u,v,\ell}\, \bar{a}^{(m)}_{u,v,\ell}}{\sum_{m=1}^{4} b^{(m)}_{u,v,\ell}},$$
$$\text{defined whenever } \sum_{m=1}^{4} b^{(m)}_{u,v,\ell} \geq 2. \tag{10}$$

This is the final per-edge, per-lag coefficient used downstream in SCM mixing. Subject to the additional constraint $u \neq v$ that removes self-links from downstream SCM mixing, we then collect all retained lags for each parent-child pair $(u, v)$ into the admissible lag set

$$\mathcal{L}_{u \to v} = \left\{ \ell \in \{0, 1, \ldots, L_{\max}\} : \sum_{m=1}^{4} b^{(m)}_{u,v,\ell} \geq 2 \right\}. \tag{11}$$

The resulting consensus graph therefore allows both instantaneous ($\ell = 0$) and delayed ($\ell > 0$) cross-variate relations, while leaving self-temporal structure to the periodic template and residual components described in Section 3.

## D. Ablation Studies

To keep the main paper compact, we collect the full variance, geometric, spectral, residual, and SCM-mixing ablations below.

### D.1. Variance Explained by Covariate

*Table 6.* Variance explained (VE, %) for each dataset and each available channel, together with the per-dataset average across available channels. Higher values indicate that the phase-aligned template explains a larger fraction of the observed signal variance.

| Dataset | Covariate 1 | Covariate 2 | Covariate 3 | Covariate 4 | Covariate 5 | Covariate 6 | Average |
|---|---|---|---|---|---|---|---|
| Bear | 57.3 | – | – | – | – | – | 57.3 |
| Panther | 39.7 | – | – | – | – | – | 39.7 |
| Azure VM | 9.1 | 25.0 | 14.5 | – | – | – | 16.2 |
| Borg | 59.3 | 15.2 | 14.8 | 0.7 | 35.4 | 59.4 | 30.8 |
| PEMS-04 | 91.9 | 84.7 | 65.4 | – | – | – | 80.7 |
| PEMS-08 | 91.2 | 84.7 | 65.1 | – | – | – | 80.3 |
| Bull | 14.8 | 0.9 | 3.4 | 2.1 | – | – | 5.3 |
| Hog | 3.2 | 0.9 | 0.3 | 1.6 | 8.8 | 1.7 | 2.8 |
| Subseasonal | 30.3 | 97.1 | 97.0 | 97.1 | – | – | 80.4 |
| Subseasonal Prec. | 32.6 | – | – | – | – | – | 32.6 |
| Res. PV | 88.7 | 88.3 | 88.8 | – | – | – | 88.6 |
| Res. Load | 88.1 | 92.0 | 93.6 | – | – | – | 91.2 |

Table 6 reports the variance explained (VE) by the phase-aligned periodic template for every dataset and channel. The values span a wide range, from below 1% on individual Hog and Bull channels to above 90% on PEMS and the residential benchmarks, reflecting genuine heterogeneity in how strongly periodic structure dominates across domains. Three broad tiers are visible. High-VE datasets, namely PEMS-04, PEMS-08, Residential PV Power, Residential Load Power, and the non-precipitation Subseasonal channels, have averages above 80%, meaning the template captures most of the signal variance before residual modelling. Moderate-VE datasets, including Bear, Panther, Subseasonal Precipitation, and Borg, sit between roughly 30% and 60% on average, so the template provides a meaningful but incomplete scaffold. Low-VE datasets, namely Azure VM, Bull, and Hog, have averages below 20%, with several individual channels near zero, indicating that periodic structure explains little of the observed variability and the residual model must carry most of the generative burden. These tiers recur as an organizing principle throughout the later ablations: template quality, as measured by VE, consistently predicts residual importance, spectral alignment, and the conditions under which SCM mixing helps or degrades. We report VE in percentage points. Columns *Covariate 1–Covariate 6* denote the ordered channels used for each dataset; datasets with fewer than six channels use '–' in the remaining entries.

## D.2. t-SNE Ablation

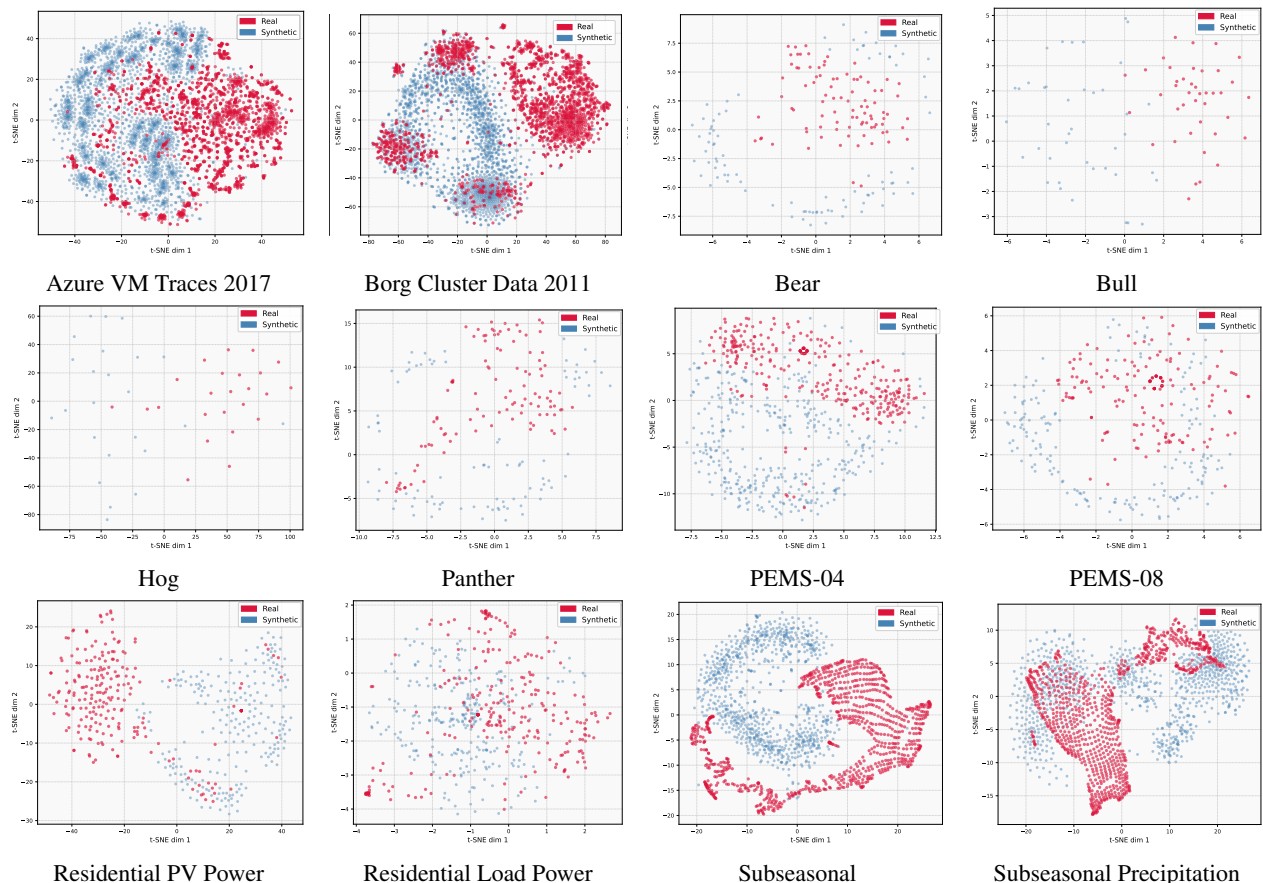

*Figure 2.* t-SNE projections of real and synthetic samples for twelve representative datasets, illustrating broad geometric alignment together with partial separation in lower-density regions.

As a qualitative diagnostic, Figure 2 offers a complementary geometric view of the same phenomenon. Across all twelve datasets, the real and synthetic point clouds remain close in the embedding space without becoming fully superposed, suggesting substantial alignment at the level of coarse support while preserving some separation between the two distributions. The absence of exact overlap is arguably favorable, since a synthetic sample that simply collapsed onto the densest empirical regions of the real data would provide comparatively limited additional coverage.

This pattern is particularly visible in the larger datasets, Azure VM Traces 2017 and Borg Cluster Data 2011. In Azure, the synthetic points extend along the outer envelope and lower-density arcs of the cloud rather than concentrating only in the most populated real regions. In Borg, they occupy several bridging and peripheral regions around the main lobes, again indicating coverage of areas that appear relatively sparse in the observed sample. For the smaller datasets such as Bear, Bull, Hog, and Panther, the two clouds more often come into contact near boundary or transition regions while remaining only partially overlapping overall.

Given the reduced sample size and the inherent variability of two-dimensional embeddings, this degree of separation is compatible with the view that the synthetic distribution tracks the same broad geometry while allocating somewhat greater mass to nearby regions that are underrepresented in the real data.

Residential PV Power is the clearest exception, where the separation is more pronounced. The synthetic cloud forms a distinct island anchored around a small subset of real points rather than spreading across the full real manifold. Because the periodic template accounts for 88.6% of signal variance in this dataset, the t-SNE embedding is driven mostly by residual variation, since the shared periodic structure contributes little to point separation. The over-dispersed residuals produced by the elevated temperature range are therefore more visible here than in other datasets, where residual variation is a smaller fraction of the total signal. This also explains why the forecasting impact remains moderate despite the geometric separation: the template carries most of the predictive signal, leaving relatively little room for the miscalibrated residual to degrade downstream perfor-

mance. The PV result points to the temperature range as a dataset-specific hyperparameter that would benefit from tuning in domains where the residual component is especially sensitive to sampling stochasticity.

## D.3. Frequency-Domain Ablation

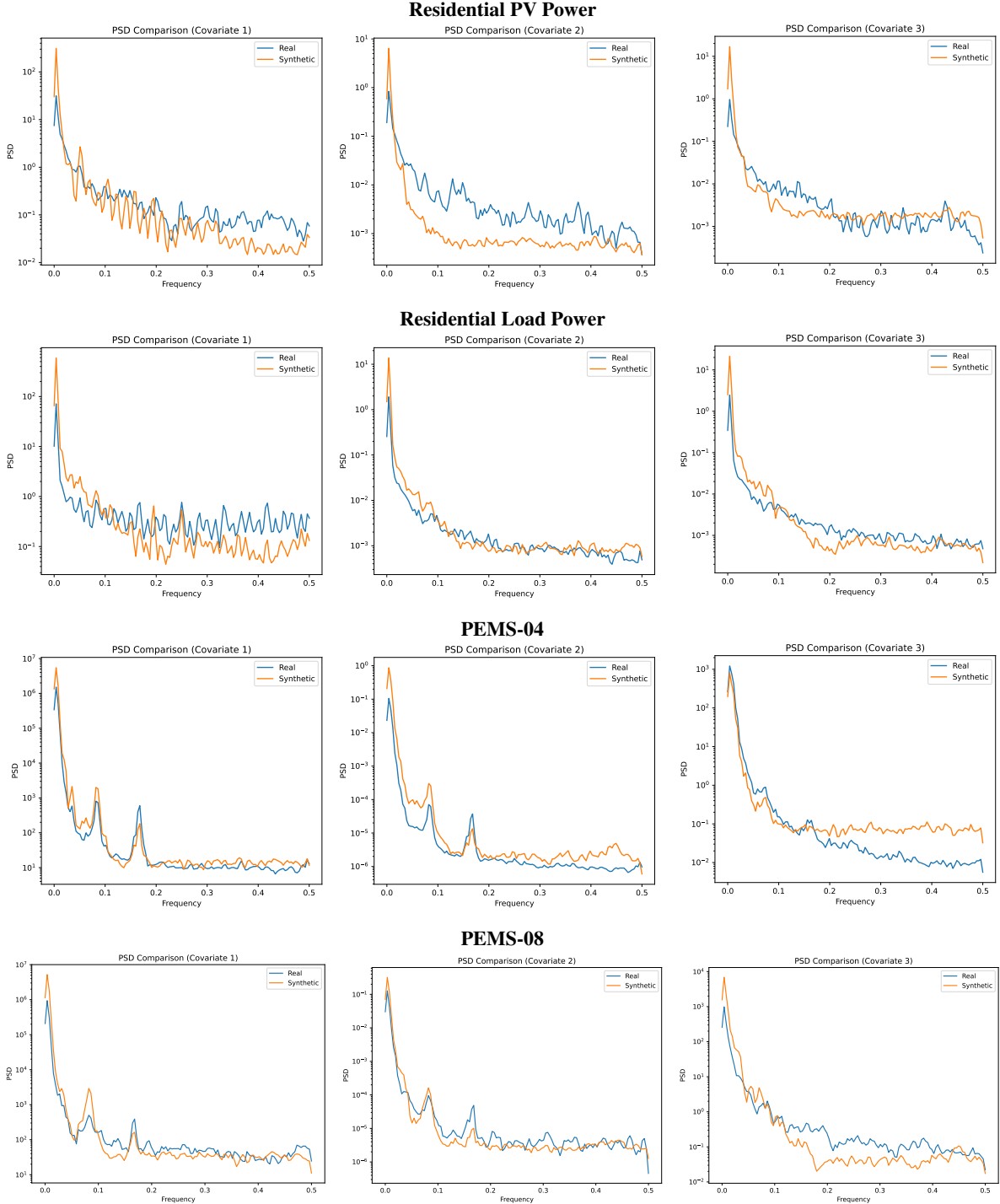

*Figure 3.* Power spectral density (PSD) comparison between real and synthetic time series across representative datasets and covariates. Each subplot shows frequency on the x-axis and spectral power density on the y-axis, with the real series shown in blue and the synthetic series shown in orange. The panels are arranged dataset-by-dataset with at most three subplots per row, except for BDG-2 Bull, which is shown with four panels in one row, and Borg Cluster Data 2011, which spans two rows. Overall, the synthetic series preserve the dominant spectral peaks and low-frequency decay patterns of the real data while allowing controlled deviations across datasets and covariates.

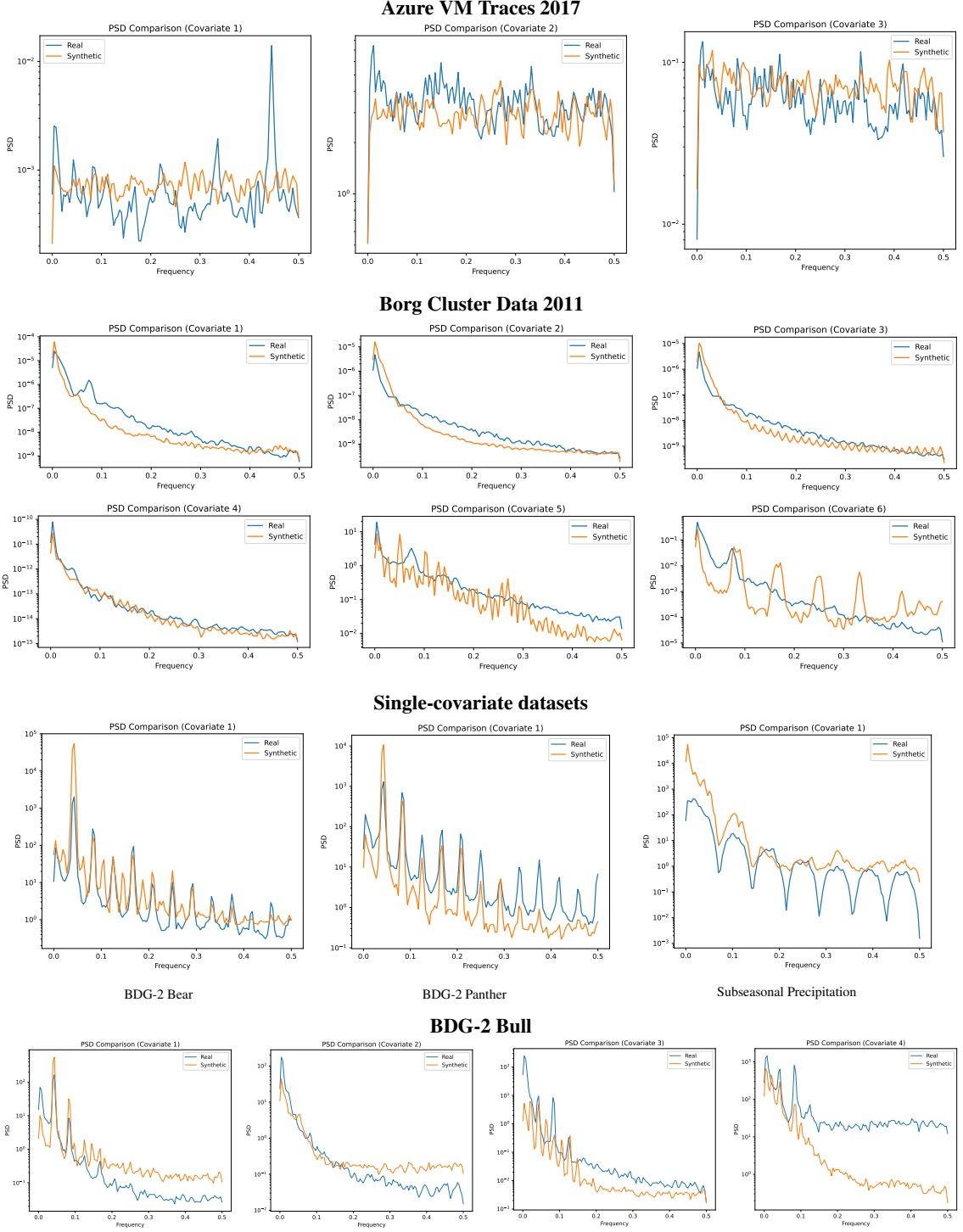

To assess spectral fidelity directly, Figure 3 provides a complementary power-spectral comparison.

The PSD comparisons in Figure 3 provide a complementary spectral view of the fidelity story told by the forecasting results. Across the included datasets, the synthetic series preserve the dominant spectral structure of the real data well, and the cases where alignment is imperfect can be traced to specific structural properties of the target domain rather than to a general failure of the pipeline.

The clearest successes are the residential energy datasets and the PEMS traffic benchmarks. For Residential PV Power and Residential Load Power (avg. VE $> 88\%$), the synthetic PSD tracks the real spectral envelope closely across all three covariates in both datasets. The dominant low-frequency decay, harmonic positions, and relative inter-peak power levels are all reproduced well. PEMS-04 and PEMS-08 (avg. VE $> 80\%$) show the same pattern in a different domain, with tight synthetic-to-real alignment across all three covariates, harmonic peaks at the correct positions, and a spectral floor that stays close to the real one. The fact that this holds for both members of the sibling pair points to genuine structural preservation rather than accidental matching. BDG-2 Bear and BDG-2 Panther, despite more moderate VE values (57.3% and 39.7%), also align well. The synthetic PSD reproduces the overall spectral decay and main harmonic positions with only minor excess power at isolated frequencies, indicating that even when the template is not dominant, the DKL residual still learns a spectrally compatible stochastic process rather than introducing spurious structure.

Two cases warrant closer attention. In Borg Cluster Data 2011, the synthetic PSD decays faster than the real one across several covariates, most visibly on covariate 4 (VE 0.7%). With 10,000 series, averaging to compute $\bar{r}$ suppresses high-frequency variability aggressively. Individual-series noise cancels in the mean, and the DKL model then learns a smoother residual process than any individual real series exhibits, causing generated residuals to underrepresent the elevated high-frequency floor of the real corpus. Azure VM Traces 2017, despite also containing 10,000 series, does not show the same pattern because its underlying signal is genuinely broadband and aperiodic. Both real and synthetic PSDs are irregular throughout, which is exactly what a faithful reproduction of a noisy process should look like. The over-smoothing effect becomes visible only when the real signal has a sustained spectral floor that the smoothed residual fails to maintain, a condition present in Borg's more structured channels but not in Azure. BDG-2 Bull, despite similarly low per-channel VE, also avoids this issue because its 41-series mean retains considerably more high-frequency structure. The Borg case is therefore specific to very large corpora with genuine mid-to-high-frequency structure, while the dominant low-frequency features remain correctly reproduced.

In Subseasonal Precipitation, the synthetic PSD preserves the seasonal harmonic positions and overall spectral shape, but the real signal has unusually sharp nulls between harmonics that the synthetic does not fully reproduce. This follows from the additive decomposition rather than from a failure of any one component. The real signal is spectrally very pure, behaving almost like a sinusoidal comb with minimal residual energy between harmonics. Once any stochastic residual is added, between-peak power necessarily increases because the residual model cannot enforce destructive interference at specific frequencies. The mismatch is therefore confined to null depth and does not affect the forecasting-relevant low- and mid-frequency envelope.

Taken together, the PSD comparisons confirm that REGEN reliably preserves the spectral structure most relevant for forecasting across the majority of the included datasets. The cases where alignment is only partial have clear structural explanations, one linked to aggressive averaging in very large corpora and the other to an unusually pure periodic signal where any additive residual raises a spectral floor that would otherwise remain nearly empty. In both cases, the dominant spectral features are still reproduced correctly.

## D.4. Residual Ablation Under TSTR

*Table 7.* Residual ablation under train-on-synthetic, test-on-real transfer (TSTR), reported only for iTransformer. The TSTR baseline values are copied from Table 1; the no-residual columns are provided to show the degradation when residual modelling is removed. Lower MSE and MAE are better.

| Dataset | TSTR | | Without Residual | | Δ | |
|---|---|---|---|---|---|---|
| | MSE↓ | MAE↓ | MSE↓ | MAE↓ | MSE | MAE |
| BDG-2 Bear | 0.41 | 0.43 | 0.45 | 0.46 | +0.04 | +0.03 |
| BDG-2 Panther | 0.36 | 0.38 | 0.39 | 0.42 | +0.03 | +0.04 |
| Azure VM Traces 2017 | 0.90 | 0.45 | 1.02 | 0.49 | +0.12 | +0.04 |
| Borg Cluster Data 2011 | 0.59 | 0.50 | 0.67 | 0.55 | +0.08 | +0.05 |
| PEMS-04 | 0.37 | 0.38 | 0.40 | 0.41 | +0.03 | +0.03 |
| PEMS-08 | 0.30 | 0.29 | 0.31 | 0.30 | +0.01 | +0.01 |
| BDG-2 Bull | 0.37 | 0.40 | 0.52 | 0.59 | +0.15 | +0.19 |
| BDG-2 Hog | 0.50 | 0.48 | 0.58 | 0.58 | +0.08 | +0.10 |
| Subseasonal | 0.40 | 0.43 | 0.41 | 0.43 | +0.01 | +0.00 |
| Subseasonal Precipitation | 1.01 | 0.73 | 1.14 | 0.80 | +0.13 | +0.07 |
| Residential PV Power | 0.26 | 0.20 | 0.26 | 0.19 | +0.00 | -0.01 |
| Residential Load Power | 0.54 | 0.38 | 0.56 | 0.39 | +0.02 | +0.01 |

To isolate the contribution of residual modelling, Table 7 compares the standard TSTR setup against a variant in which synthetic data is generated without sampled residuals, reported only for iTransformer. Removing the DKL residual consistently degrades performance, but the magnitude of that degradation is strongly structured by how much signal variance the periodic template already explains.

The clearest pattern emerges when the residual-removal degradation is related back to the per-dataset variance explained (VE) values in Table 6. Datasets where the phase-aligned template captures little of the original signal variance suffer the largest collapse when residuals are removed: BDG-2 Bull (avg. VE 5.3%) degrades by +0.15 MSE and +0.19 MAE, Azure VM Traces 2017 (avg. VE 16.2%) by +0.12 MSE and +0.04 MAE, and BDG-2 Hog (avg. VE 2.8%) by +0.08 MSE and +0.10 MAE. By contrast, datasets where the template is highly explanatory are largely unaffected: PEMS-08 (avg. VE 80.3%) degrades by only +0.01 MSE and +0.01 MAE, Subseasonal (avg. VE 80.4%) by +0.01 MSE and +0.00 MAE, Residential PV Power (avg. VE 88.6%) shows no meaningful change, and Residential Load Power (avg. VE 91.2%) changes by only +0.02 MSE and +0.01 MAE.

To quantify this relationship, we compute the Spearman rank correlation between per-dataset average VE and the absolute MSE degradation upon residual removal, finding $\rho = -0.8225$. The same relationship holds for MAE degradation, with $\rho = -0.8858$. These coefficients confirm that template quality is a reliable predictor of residual importance: the less structure the template captures, the more load-bearing the residual model becomes.

This has a direct mechanistic interpretation. In high-VE settings such as PEMS, Subseasonal, and the residential PV/load benchmarks, the periodic template dominates the generative signal and the residual model acts mainly as a stochastic correction. In low-VE settings such as Bull, Hog, and Azure VM, the template fails to capture dominant variability—including irregular load spikes, bursty VM utilization, and aperiodic demand shifts—so the DKL residual must carry much more of the generative burden. In these cases, removing it erases the structured variability that makes the synthetic series informative as a training source and leaves the generator with a much more rigid view of the process. The residual model is therefore not uniformly a correction term but, conditionally, a principal generative component whose importance is determined by the degree to which the target domain exhibits stable periodic structure.

### D.5. SCM Mixing Ablation Under TSTR

*Table 8.* SCM-mixing ablation under train-on-synthetic, test-on-real transfer (TSTR), reported only for iTransformer. The TSTR baseline values are copied from Table 1; the w/o SCM mixing columns are provided to show the degradation when SCM-based mixing is removed. Lower MSE and MAE are better.

| Dataset | TSTR | | Without SCM Mixing | | Δ | |
|---|---|---|---|---|---|---|
| | MSE↓ | MAE↓ | MSE↓ | MAE↓ | MSE | MAE |
| Azure VM Traces 2017 | 0.90 | 0.45 | 0.96 | 0.46 | +0.06 | +0.01 |
| Borg Cluster Data 2011 | 0.59 | 0.50 | 0.62 | 0.54 | +0.03 | +0.04 |
| PEMS-04 | 0.37 | 0.38 | 0.38 | 0.39 | +0.01 | +0.01 |
| PEMS-08 | 0.30 | 0.29 | 0.32 | 0.32 | +0.02 | +0.03 |
| BDG-2 Bull | 0.37 | 0.40 | 0.37 | 0.42 | +0.00 | +0.02 |
| BDG-2 Hog | 0.50 | 0.48 | 0.48 | 0.45 | -0.02 | -0.03 |
| Subseasonal | 0.40 | 0.43 | 0.47 | 0.51 | +0.07 | +0.08 |
| Residential PV Power | 0.26 | 0.20 | 0.30 | 0.29 | +0.04 | +0.09 |
| Residential Load Power | 0.54 | 0.38 | 0.56 | 0.42 | +0.02 | +0.04 |

To isolate the contribution of multivariate structural coupling, Table 8 compares the standard TSTR setup against a variant in which SCM-based mixing is removed during synthetic generation. Since this ablation is meaningful only for multivariate datasets, the univariate BDG-2 Bear, BDG-2 Panther, and Subseasonal Precipitation datasets are omitted. As above, we give the model name in the caption and copy the TSTR baseline values directly from Table 1.

The SCM ablation points to a modest but mostly positive effect. Most datasets worsen slightly without SCM mixing, while BDG-2 Hog improves marginally and BDG-2 Bull is nearly unchanged. The clearest explanation is not dimensionality alone but how much evidence is available per candidate edge in the consensus graph. For Hog, the combination of few series and relatively many channels makes edge selection noisy: with $\tau_{\text{freq}} = 0.2$, an edge can survive after appearing in only 5 of the 24 series. That is a low bar, so some spurious dependencies can enter the graph and then be propagated across channels during generation. By contrast, Borg is also higher-dimensional but has vastly more series, so each candidate edge is evaluated against much stronger evidence and removing SCM mixing hurts as expected. Bull sits between these cases, with fewer channels and 41 series, which is consistent with its near-zero SCM effect. The PEMS datasets also lie in a better-supported regime and show small but consistent gains from SCM mixing. Overall, the pattern suggests that SCM mixing helps when the consensus graph is well supported by the available data and becomes close to neutral when evidence is limited, with Hog as the clearest failure case. A stricter or adaptive consensus threshold for low-data, higher-dimensional settings is therefore a natural direction for future work.

