# OpenReview forum: "ReGeN: Reference-Guided Synthetic Multivariate Time Series Generation for Forecasting"
_ICML.cc/2026/Workshop/FMSD — FMSD @ ICML 2026 Poster_

### Official Review · Reviewer_3dbx · 2026-05-21
**More recent baselines are missing**

**Rating:** 6
**Confidence:** 3

**Review:**

### Summary
The paper proposes REGEN, a pipeline to synthesize sequences for the training of time series forecasting models. The pipeline starts by decomposing real sequences into periodic structures and a stochastic part. The generated sequences are coupled using a DAG similar to CauKer. In an empirical evaluation, the authors demonstrate consistent improvements over baselines in a TRTS and TRSTR setting.

### Strengths
- Experiments are conducted on different types of architectures, i.e., a transformer, an SSM, and a linear model.
- Figure 1 provides a good overview of the model, making the proposed pipeline easy to understand.
- The empirical results of the TRTR vs TRSTR comparison look promising. Including synthetic samples seems to improve the non-linear models.

### Weaknesses
- The related work section should be extended and differentiated against. What are the differences between CauKer, and what are the specific contributions?
- It is unclear why ReGen is compared against TimeGAN. That baseline is from 2019 and is not the state-of-the-art time-series generative model, making the empirical evaluation insufficient. Diffusion-based baselines should be included. Further, the comparison against CauKer implies only marginal improvements.
- A -> A and B -> B in Table 1 are missing and would help to interpret the results.

---

### Official Review · Reviewer_VDFQ · 2026-05-21
**Clear narrative, solid contribution, with experimental results slightly weaker**

**Rating:** 8
**Confidence:** 4

**Review:**

# Summary

To solve the problem of generating synthetic data for pretraining time series foundation models, this paper introduces REGEN, a reference-guided pipeline that generates multivariate training data based on a small number of real multivariate sequences. The framework consists of several components, including a phase-aligned template, deep-kernel GP residuals, and a lagged-edge DAG for cross-variable structure. Empirical results show that REGEN preserves domain-specific multivariate structure and yields competitive downstream forecasting performance.

# Strengths

1. This paper has well-motivated decompositions. Borrowing periodic structure directly from the reference and generating novelty through the residual path is a reasonable way to balance fidelity and diversity when only a few real sequences are available. Each component is grounded in established prior work.
2. Multi-method consensus DAG estimation is more principled than single-estimator method, and removing self-edges correctly avoids double-counting the template's autoregressive structure.
3. The paper establishes experiments with a broad empirical scope and shows downstream forecasting performance with wide coverage.
4. The paper is well-written.

# Areas for Improvement

1. The paper's evaluation of synthetic data quality is mostly based on downstream utility, with no quantitative distribution metrics and only qualitative figures. For the t-SNE figures, the performance does not seem very good visually, with not much overlap and some separation between the real and synthetic distributions. Besides, the paper does not offer theoretical guarantee that synthetic samples resemble the real distribution.
2. The manually specified period and template-based design make REGEN heuristic and likely to struggle on non-periodic, event-driven, or regime-switching data.
3. If the reference data contains labels, REGEN lacks the ability to preserve them in the synthetic output.
4. The paper does not demonstrate that REGEN's benefits can scale to truly large foundation models. In the experiments, the foundation-model evaluation uses only Moirai-small.

# Detailed Comments

1. Add metrics that can measure the distributional similarity of the real and generated time series, like Wasserstein distance.
2. Add mechanisms to preserve labels and the corresponding structure information.
3. Add experiments of non-periodic/event-driven datasets.
4. Extend pretraining to larger foundation models.

# Justification of Score

REGEN makes solid methodological contributions with broad empirical support. The main weaknesses, including lack of quantitative distribution metrics, modest downstream task utility, and lack of scaling to large foundation models, are evaluation gaps that can be optimized rather than fundamental flaws, so I lean toward a clear accept for this paper.